# Balancing Accuracy, Safety, and Cost in Mediastinal Diagnostics: A Systematic Review of EBUS and Mediastinoscopy in NSCLC

**DOI:** 10.3390/healthcare13151924

**Published:** 2025-08-06

**Authors:** Serban Radu Matache, Ana Adelina Afetelor, Ancuta Mihaela Voinea, George Codrut Cosoveanu, Silviu-Mihail Dumitru, Mihai Alexe, Mihnea Orghidan, Alina Maria Smaranda, Vlad Cristian Dobrea, Alexandru Șerbănoiu, Beatrice Mahler, Cornel Florentin Savu

**Affiliations:** 1Department of Thoracic Surgery, “Carol Davila” University of Medicine and Pharmacy, 050474 Bucharest, Romania; radu.matache@gmail.com (S.R.M.); anca1502@gmail.com (A.M.V.); alina.smaranda@drd.umfcd.ro (A.M.S.); dobreavladcristian@gmail.com (V.C.D.); drsavu25@yahoo.com (C.F.S.); 2“Marius Nasta” Institute of Pneumophtisiology, 050159 Bucharest, Romania; alexe.mihai@yahoo.com (M.A.); mihnea.org@gmail.com (M.O.); beatrice.mahler@umfcd.ro (B.M.); 3Ponderas Academic Hospital, 014142 Bucharest, Romania; codrut.cosoveanu@gmail.com; 4Emergency Clinical Hospital “Prof. Dr. Agrippa Ionescu”, 011356 Bucharest, Romania; dumitrusm@gmail.com; 5Bucharest University Emergency Hospital, 050098 Bucharest, Romania; alexandru_serbanoiu@yahoo.com; 6Department of Radiology, “Carol Davila” University of Medicine and Pharmacy, 050474 Bucharest, Romania; 7Department of Pneumology, “Carol Davila” University of Medicine and Pharmacy, 050474 Bucharest, Romania

**Keywords:** EBUS-TBNA, mediastinoscopy, surgery, flexible bronchoscopy, mediastinal staging, mediastinal imaging, non-small lung cancer

## Abstract

**Background:** Mediastinal staging plays a critical role in guiding treatment decisions for non-small cell lung cancer (NSCLC). While mediastinoscopy has been the gold standard for assessing mediastinal lymph node involvement, endobronchial ultrasound-guided transbronchial needle aspiration (EBUS-TBNA) has emerged as a minimally invasive alternative with comparable diagnostic accuracy. This systematic review evaluates the diagnostic performance, safety, cost-effectiveness, and feasibility of EBUS-TBNA versus mediastinoscopy for mediastinal staging. **Methods:** A systematic literature review was conducted in accordance with PRISMA guidelines, including searches in Medline, Scopus, EMBASE, and Cochrane databases for studies published from 2010 onwards. A total of 1542 studies were identified, and after removing duplicates and applying eligibility criteria, 100 studies were included for detailed analysis. The extracted data focused on sensitivity, specificity, complications, economic impact, and patient outcomes. **Results:** EBUS-TBNA demonstrated high sensitivity (85–94%) and specificity (~100%), making it an effective first-line modality for NSCLC staging. Mediastinoscopy remained highly specific (~100%) but exhibited slightly lower sensitivity (86–90%). EBUS-TBNA had a lower complication rate (~2%) and was more cost-effective, while mediastinoscopy provided larger biopsy samples, essential for molecular and histological analyses. The need for general anaesthesia, longer hospital stays, and increased procedural costs make mediastinoscopy less favourable as an initial approach. Combining both techniques in select cases enhanced overall staging accuracy, reducing false negatives and improving diagnostic confidence. **Conclusions:** EBUS-TBNA has become the preferred first-line mediastinal staging method due to its minimally invasive approach, high diagnostic accuracy, and lower cost. However, mediastinoscopy remains crucial in cases requiring posterior mediastinal node assessment or larger tissue samples. The integration of both techniques in a stepwise diagnostic strategy offers the highest accuracy while minimizing risks and costs. Given the lower hospitalization rates and economic benefits associated with EBUS-TBNA, its widespread adoption may contribute to more efficient resource utilization in healthcare systems.

## 1. Introduction

The accurate evaluation of mediastinal lymph nodes plays a pivotal role in diagnosing, staging, and managing thoracic diseases, particularly lung cancer. This process concludes with critical treatment decisions, including surgical resecability, chemotherapy and radiation therapy, and directly impacts patient prognosis. Historically, mediastinoscopy has been considered the gold standard for mediastinal lymph node assessment due to its well-established diagnostic accuracy, high sensitivity, and specificity. However, this procedure is invasive, requiring general anaesthesia and hospitalization, and carries risks of complications such as bleeding, infection, and, in rare cases, major vascular injury [1].

In recent years, endobronchial ultrasound (EBUS)-guided transbronchial needle aspiration has gained prominence as a minimally invasive alternative. EBUS allows for the real-time ultrasound-guided sampling of mediastinal and hilar lymph nodes via the tracheobronchial wall, often performed on an outpatient basis with a moderate sedation of local anaesthesia alone. The potential benefits of EBUS include reduced hospitalization costs, fewer complications, and the ability to repeat the procedure when necessary. However, the extent to which EBUS matches mediastinoscopy in diagnostic accuracy, sensitivity, and specificity remains an area of active investigation. Moreover, questions persist regarding the relative safety of each technique, the cost implications (including both procedural and hospitalization costs), and the time required for the pathological analysis of biopsy specimens obtained via each method [2].

Another critical consideration is the potential for technique repetition. While mediastinoscopy is often limited by surgical scarring and anatomical disruption, EBUS is more amenable to repetition in cases of diagnostic uncertainty or disease recurrence. Additionally, differences in the time required for pathology processing between these techniques may have downstream effects on clinical decision-making and overall patient management timelines.

This systematic review aims to comprehensively compare EBUS and mediastinoscopy across key domains, including diagnostic accuracy, sensitivity, specificity, safety, complication rates, cost-effectiveness (overall and hospitalization costs), the feasibility of technique repetition and anaesthesia and patient care. By synthesizing and analysing the available evidence, this review seeks to provide a detailed understanding of the strengths, limitations, and clinical applications of EBUS and mediastinoscopy, ultimately guiding healthcare providers in choosing the most appropriate diagnostic approach for their patients.

## 2. Materials and Methods

The present systematic review was conducted in accordance with the Preferred Reporting Items for Systematic Reviews and Meta-Analyses recommendations. We focused on studies analysing lung cancer patients who underwent diagnosis interventions such as either EBUS-TBNA or mediastinoscopy with the objective of comparing the two procedures regarding diagnostic accuracy, sensitivity, specificity, safety, complication rates, cost-effectiveness and the feasibility of technique repetition.

### 2.1. Information Source and Research Technique

Systematic searches were conducted in the Medline, Scopus, EMBASE and Cochrane databases, the data ranges being set starting with 01.01.2010. We also performed grey literature research, which included references in the selected articles. The following search strategy was used for each database: “lung cancer” OR “lung carcinoma” OR “pulmonary cancer” OR “NSCLC” OR “cancer of the lung” OR “lung neoplasm” AND “EBUS” OR “EBUS-TBNA” OR “Endobronchial ultrasound-guided transbronchial needle aspiration” OR “endobronchial ultrasound” AND “mediastinoscopy”.

### 2.2. Eligibility Criteria

The studies selected consisted of randomized and nonrandomized clinical trials, comparative studies, scientific reviews, observational studies, multicentre studies and retrospective studies.

The types of interventions studied were EBUS-TBNA and mediastinoscopy. The standard for comparing the two methods was the result of tumour resection surgery, based on systematic mediastinal lymph node sampling or dissection.

### 2.3. Study Selection

After performing the initial search on each database, we received a total number of 1542 results. We then performed automatic duplicate removal and 66 duplicates identified by Covidence, thus leaving us with 1445 references. All studies were screened manually by title and abstract first; thus, we excluded 1023 studies for reasons including irrelevant population, inappropriate study design, the lack of relevant intervention or outcome and conference abstracts without full data. The remaining 422 full-text studies were assessed for eligibility, out of which 89 studies were not fully retrieved and 322 studies were excluded. Of the 89 full-text articles that could not be retrieved, the most common reasons included a lack of access to subscription-based journals, the unavailability of full texts beyond abstracts and unsuccessful author contact. The main reasons for exclusion were wrong study design and wrong comparison terms. A risk of bias assessment was conducted independently by two reviewers, who screened the abstracts and full-text articles separately. Discrepancies were resolved through discussion or consultation with a third reviewer (Figure 1).

### 2.4. Synthesis of Results and Analysis

The 100 studies included in the systematic review were analysed directly on the Covidence platform, following the same data extraction template. After finalizing the data extraction, the extracted data were systematically reviewed and verified for accuracy and consistency. Each study’s relevant information, including study design, sample size, population characteristics, interventions, comparators, outcomes, and results, was documented in the standardized template.

Once the extraction was completed, discrepancies between reviewers were resolved through discussion or consultation with a third reviewer, ensuring the reliability of the data. The cleaned dataset was then exported for further analysis, including the identification of trends, synthesis of findings, and preparation of summary tables for the Results Section of the systematic review. This structured process ensured that all studies were uniformly assessed and their contributions to the review were accurately represented.

## 3. Results

### 3.1. Sensitivity and Specificity of EBUS-TBNA

EBUS-TBNA has emerged as a minimally invasive technique offering the real-time ultrasound-guided sampling of mediastinal and hilar lymph nodes. Several studies have demonstrated its high sensitivity and specificity, particularly in NSCLC staging.

EBUS-TBNA exhibits sensitivity ranging from 85% to 94% for detecting mediastinal lymph node metastases. In a prospective study by Yasufuku et al., sensitivity was reported at 94%, with a specificity of 100%, underscoring its diagnostic precision [3]. Another systematic review by Kokkonouzis et al. found a pooled sensitivity of 87%, highlighting its reliability across different settings [4]. Meanwhile, De Dominicis et al. stated a lower sensitivity of 83.3% in their study, which was published in 2015, but the patient pool was lower than other studies reviewed [5]. However, sensitivity can vary depending on factors such as operator experience, lymph node size, and the number of nodes sampled [6].

EBUS-TBNA consistently achieves specificity approaching 100%, indicating a high ability to confirm malignancy when detected cytologically [7]. The integration of real-time ultrasound guidance allows for the precise localization of lymph nodes, even those that are small (<1 cm) or in challenging locations. This feature enhances the diagnostic accuracy and reduces sampling errors [8]. The low false-positive rate of EBUS-TBNA is particularly advantageous in avoiding overtreatment.

### 3.2. Sensitivity and Specificity of Carlens Mediastinoscopy

Mediastinoscopy has traditionally been regarded as the gold standard for mediastinal staging due to its ability to provide thr direct visualization and biopsy of lymph nodes. The sensitivity of mediastinoscopy for detecting metastatic lymph nodes ranges from 86% to 90%. A meta-analysis by Detterbeck et al. confirmed its reliability, with pooled sensitivity of 89% across studies, which was restated by Osarogiagbon et al. in 2019 [9,10]. Sensitivity is particularly high for central mediastinal nodes (stations 2R, 2L, 4R, 4L, and 7).

Mediastinoscopy, like EBUS-TBNA, achieves near-perfect specificity (99–100%), ensuring the accurate exclusion of malignancy when lymph node biopsies are negative [11].

Table 1. Comparative Analysis of EBUS-TBNA and Mediastinoscopy in NSCLC Diagnosis presents a chronological comparison of ten studies published between 2009 and 2020, evaluating the diagnostic performance of EBUS-TBNA and mediastinoscopy in the mediastinal staging of NSCLC. Key metrics such as sensitivity, specificity, and the number of procedures for each modality are highlighted, offering insights into their respective accuracies and clinical applications over time.

### 3.3. Comparison Between EBUS-TBNA and Mediastinoscopy Regarding Sensitivity and Specificity

As a bronchoscopy procedure, EBUS-TBNA is associated with fewer complications, faster recovery, and suitability for high-risk patients, but its sensitivity is highly dependent on the skill and experience of the bronchoscopist [11,20]. Smaller biopsy specimens obtained through EBUS-TBNA may be insufficient for advanced molecular testing in some cases. EBUS-TBNA is particularly effective for sampling nodes in stations 10R and 10L, which are often inaccessible during mediastinoscopy [21].

### 3.4. Diagnostic Yield and Sample Adequacy

Accurate mediastinal staging in NSCLC depends on obtaining high-quality samples for both diagnosis and molecular analysis. EBUS-TBNA and mediastinoscopy are pivotal techniques in this process, each with distinct strengths. EBUS-TBNA, a minimally invasive procedure, offers comparable diagnostic yield and sample adequacy to mediastinoscopy, while providing the added benefit of being an outpatient procedure with fewer complications. Mediastinoscopy, on the other hand, remains valuable for accessing certain nodal stations and obtaining larger tissue samples.

Table 2. Diagnostic Yield, Sample Adequacy, and Molecular Testing Feasibility of EBUS-TBNA in NSCLC provides a comprehensive summary of studies that compare EBUS-TBNA to mediastinoscopy in terms of diagnostic yield, sample adequacy, and molecular testing feasibility for patients with NSCLC. These studies highlight the ability of EBUS-TBNA to obtain high-quality samples suitable for both histopathological evaluation and advanced molecular analysis.

### 3.5. Risks and Complications

Table 3. Comparison of Risks and Complications Associated with EBUS-TBNA and Mediastinoscopy provides a comprehensive comparison of the risks and complications associated with EBUS-TBNA and mediastinoscopy and outlines common risks, rare complications, morbidity rates, and mortality rates, highlighting the contrasting safety profiles of these techniques. This comparison underscores the minimally invasive nature and lower risk profile of EBUS-TBNA, which makes it the preferred option for many patients, while also detailing the higher risks associated with mediastinoscopy due to its surgical approach [31,32,33].

One of the most reported complications of EBUS-TBNA is minor bleeding, occurring in approximately 1–2% of cases. This bleeding is generally minor, self-limiting, and localized to the puncture site of the sampled lymph nodes. The use of real-time imaging enables operators to promptly identify and address any bleeding, ensuring patient safety during the procedure. Transient hypoxemia, often associated with sedation or bronchoscopic manipulation, is another potential complication [18]. However, this condition typically resolves quickly with supplemental oxygen and rarely causes significant clinical issues. In rare instances, trauma to the vocal cords may occur during bronchoscope insertion, leading to temporary hoarseness or discomfort, but these cases are infrequent and generally mild [34].

Serious complications associated with EBUS-TBNA are exceedingly rare. Pneumothorax occurs in less than 1% of cases and is primarily linked to the inadvertent needle puncture of the pleura during sampling. This risk is mitigated by the careful visualization of anatomical structures provided by the ultrasound guidance system. Similarly, post-procedural infections, including mediastinitis, are rare, with reported rates below 0.5%. Adherence to stringent sterilization protocols and aseptic techniques significantly minimizes this risk [35,36].

Despite its effectiveness, mediastinoscopy is associated with a higher risk profile due to its invasive nature, necessitating general anaesthesia and a surgical setting. One of the most significant risks associated with mediastinoscopy is bleeding, occurring in up to 1% of cases. Haemorrhage is typically associated with the biopsy of vascularized lymph nodes and may require prompt surgical intervention to manage [36].

Pneumothorax is another potential complication, with an incidence of 1–3%, resulting from inadvertent pleural puncture during the procedure. This condition often requires chest tube placement and close monitoring for resolution [37]. Injury to the recurrent laryngeal nerve is a notable risk, with an occurrence rate of 1–2%. Such injuries can lead to unilateral or bilateral vocal cord paralysis, manifesting as hoarseness, dysphonia, or, in severe cases, respiratory compromise. While rare, infections, including wound infections and mediastinitis, are reported in less than 0.1% of cases. Proper sterile techniques and perioperative antibiotic administration significantly reduce the likelihood of such complications [38,39,40].

The overall morbidity rate of mediastinoscopy ranges from 3 to 5%, reflecting its surgical nature and the requirement for general anaesthesia. Despite this, the procedure has a very low mortality rate, estimated at less than 0.1%, making it relatively safe for most patients when performed by experienced surgeons in well-equipped settings [11]. Postoperative recovery is longer than that of minimally invasive procedures, with patients commonly reporting pain, fatigue, or difficulty swallowing during the initial days following surgery. Hospital stays are typically brief but required in many cases to monitor for complications [41,42].

Table 4. Comparative Analysis of EBUS-TBNA and Mediastinoscopy—complications provides a detailed comparison of studies published from 2018 onward that evaluate the diagnostic performance and safety profiles of EBUS-TBNA and mediastinoscopy in the mediastinal staging of NSCLC. The table highlights key metrics, including the number of procedures performed, complication rates, mortality rates, and the incidence of adverse events associated with each technique. This comparison underscores the distinct safety and risk profiles of these procedures, offering valuable insights into their clinical applications.

### 3.6. Cost-Effectiveness

The cost-effectiveness of diagnostic methods for mediastinal staging in NSCLC is a key factor influencing their adoption. EBUS-TBNA and mediastinoscopy differ significantly in terms of cost, recovery, and complication rates. With healthcare systems worldwide constrained by budgetary limits, evaluating the economic impact of these procedures is essential for selecting the most efficient and effective approach [50,51,52,53,54].

Table 5. Cost-Effectiveness of EBUS-TBNA and Mediastinoscopy in Mediastinal Staging of NSCLC provides a comprehensive overview of studies evaluating the cost-effectiveness of EBUS-TBNA compared to mediastinoscopy for mediastinal staging in non-small cell lung cancer (NSCLC). The table highlights the procedures analysed, overall cost per procedure, and the geographical context of each study. These findings consistently demonstrate that EBUS-TBNA offers significant cost savings while maintaining comparable diagnostic efficacy.

Regarding outpatient feasibility, EBUS-TBNA is typically performed under moderate sedation in an outpatient setting, reducing hospitalization costs. In contrast, mediastinoscopy requires general anaesthesia, an operating room, and postoperative monitoring, significantly increasing costs [60]. The direct costs of EBUS-TBNA are generally lower than those of mediastinoscopy. A study by Harewood et al. estimated that the per-patient cost of EBUS-TBNA in the United States is approximately USD 2800, compared to USD 6500 for mediastinoscopy [61]. Similar trends have been observed in European settings, where EBUS-TBNA is consistently less expensive than surgical staging [62].

The lower complication rate of EBUS-TBNA translates into fewer additional healthcare expenditures. Complications such as minor bleeding or transient hypoxemia are easily managed, whereas mediastinoscopy-related complications (e.g., pneumothorax, nerve injury) often require extended hospital stays or additional interventions [63].

### 3.7. Anaesthesia and Patient Care

Optimizing anaesthetic techniques in EBUS-TBNA and mediastinoscopy is crucial for enhancing patient safety, procedural success, and diagnostic accuracy while minimizing complications and recovery time [64].

Both endobronchial ultrasound-guided transbronchial needle aspiration (EBUS-TBNA) and mediastinoscopy are widely used techniques for this purpose, with anaesthesia considerations being a key factor in procedural success, patient safety, and overall efficiency. The choice of anaesthesia—ranging from local anaesthesia only, conscious sedation to general anaesthesia—can significantly impact diagnostic yield, patient tolerance, recovery time, and healthcare costs [65].

EBUS-TBNA is commonly performed under moderate sedation (conscious sedation) using agents such as midazolam, propofol, or dexmedetomidine, often combined with opioid analgesics like fentanyl or meperidine. This approach allows for rapid recovery, minimizes complications, and is generally well-tolerated. However, general anaesthesia is also employed in some cases to optimize patient comfort, reduce movement, and potentially improve diagnostic accuracy by allowing for more precise sampling. Despite these benefits, general anaesthesia may increase procedural costs, prolong recovery time, and pose risks related to airway management. The most important facts to take into consideration regarding anaesthesiology procedure is the patients’ general health status and other simultaneous afflictions [65,66].

Conversely, mediastinoscopy is an inherently more invasive surgical procedure that mandates general anaesthesia due to its requirement for direct mediastinal dissection and tissue biopsy. Although mediastinoscopy remains a gold standard for assessing certain lymph node stations, it is associated with higher complication rates, longer recovery periods, and increased anaesthesia-related risks compared to EBUS-TBNA [65,67].

Table 6. Anaesthesia Practices in EBUS-TBNA and Mediastinoscopy presents recent studies that examine various anaesthesia strategies in EBUS-TBNA and mediastinoscopy, comparing their impact on procedural success, safety, and patient outcomes. The data provide insight into the ongoing evolution of sedation and anaesthesia protocols in these diagnostic techniques, highlighting their implications for clinical practice.

## 4. Discussion

The comparative evaluation of EBUS-TBNA and mediastinoscopy in the mediastinal staging of non-small cell lung cancer (NSCLC) underscores the distinct advantages and limitations of each technique. As lung cancer treatment continues to advance, the role of accurate and minimally invasive diagnostic modalities has become increasingly significant. EBUS-TBNA has emerged as a preferred first-line diagnostic tool, offering a combination of high diagnostic yield, safety, and cost-effectiveness, which makes it a suitable option in diverse clinical settings [75,76,77].

EBUS-TBNA facilitates the precise sampling of mediastinal lymph nodes under real-time ultrasound guidance, which improves targeting accuracy and minimizes complications. Numerous studies have demonstrated that the diagnostic accuracy of EBUS-TBNA is comparable to mediastinoscopy, particularly in the evaluation of central mediastinal lymph nodes (stations 2R, 2L, 4R, 4L, and 7) [78,79]. Furthermore, EBUS-TBNA has demonstrated excellent sample adequacy, often providing sufficient material for cytological, histological, and molecular testing, which is crucial in the era of personalized lung cancer therapy. Given the increasing reliance on genetic and biomarker testing (EGFR, ALK, ROS1, PD-L1, and KRAS mutations) for targeted treatments, the ability to perform molecular analysis on minimally invasive samples represents a significant advancement in lung cancer care [2,80,81,82].

Mediastinoscopy, however, continues to hold value in specific cases. While EBUS-TBNA is preferred for its ability to provide real-time visualization and precise sampling, mediastinoscopy remains the gold standard when larger tissue biopsies are required for comprehensive molecular and histopathological analysis. Additionally, mediastinoscopy enables access to posterior mediastinal lymph nodes, which may not be readily accessible with EBUS-TBNA. However, the higher complication rates, prolonged hospital stays, and increased overall costs associated with mediastinoscopy have led to a gradual decline in its routine use. Complications such as pneumothorax, bleeding, recurrent laryngeal nerve injury, and wound infections are more frequently reported compared to EBUS-TBNA. The necessity of general anaesthesia and surgical expertise further restricts its widespread applicability, particularly in centres with limited thoracic surgical resources [27,83,84,85].

From an economic standpoint, multiple cost-effectiveness studies suggest that EBUS-TBNA is a more resource-efficient approach due to its ability to be performed in an outpatient setting and its lower complication rates. The reduced need for hospitalization and post-procedural monitoring further contributes to cost savings in healthcare systems with limited resources. In clinical practice, EBUS-TBNA is increasingly used as the first-line modality for mediastinal staging due to its minimally invasive nature and high diagnostic yield. Mediastinoscopy is now reserved for cases where EBUS-TBNA results are inconclusive or when access to specific nodal stations is required. Nevertheless, mediastinoscopy continues to play a crucial role in select cases, particularly when EBUS-TBNA is insufficient or when a more invasive surgical approach is needed to guide treatment decisions [11,86,87].

Recent studies have demonstrated that combining EBUS-TBNA with mediastinoscopy significantly improves overall diagnostic yield. The European Society of Thoracic Surgeons (ESTS) guidelines now recommend a multimodal approach to optimize staging accuracy, particularly in cases with a high clinical suspicion of mediastinal lymph node metastases despite negative EBUS-TBNA findings [88,89].

A combined approach is particularly beneficial in the following scenarios:When EBUS-TBNA results are negative, but PET-CT or clinical findings suggest a high pretest probability of mediastinal disease [90,91].When posterior mediastinal lymph nodes (stations 5, 6, 8, and 9) require evaluation, which is beyond the reach of EBUS-TBNA. However, in this case, association with EUS-TBNA or lymph node biopsy by left VATS approach is necessary [92].In cases where larger tissue samples are necessary for complex molecular testing and immunohistochemistry [93,94].In post-treatment restaging, particularly in patients with previous radiation therapy or chemotherapy, where lymph node fibrosis may reduce EBUS-TBNA sample adequacy [20,95].

Several studies have reported that adding mediastinoscopy after an initial negative EBUS-TBNA result increases overall sensitivity for detecting mediastinal metastases by 10–20%, reducing false-negative rates and improving treatment decision-making. However, a sequential strategy is often preferred over a simultaneous combination, as most patients can be staged adequately with EBUS-TBNA alone, and mediastinoscopy should be reserved for cases where additional clarification is required [96,97].

Looking ahead, technological advancements are poised to further refine the precision, diagnostic yield, and utility of EBUS-TBNA. The integration of robotic-assisted bronchoscopy, artificial intelligence-driven image analysis, and advanced biopsy needle designs could improve tissue acquisition rates and sample adequacy, particularly in challenging cases with small or difficult-to-reach lymph nodes. Additionally, hybrid staging strategies combining EBUS-TBNA with confirmatory mediastinoscopy or surgical lymphadenectomy may optimize overall diagnostic accuracy while ensuring minimized invasiveness and improved patient safety [1,98,99].

One limitation of this systematic review is the variability in study methodologies, operator expertise, and patient selection criteria, which may introduce heterogeneity in reported diagnostic performance and complication rates. Additionally, not all studies included in this review were freely available in full text, which may have limited the depth of analysis for certain findings. While this review provides a comprehensive comparison of EBUS-TBNA and mediastinoscopy, it does not account for emerging technologies such as robotic-assisted bronchoscopy or AI-driven image analysis, which may further refine mediastinal staging accuracy. Future research should focus on long-term patient outcomes, the impact of combined diagnostic approaches on survival rates, and cost-effectiveness analyses in different healthcare systems, particularly in resource-limited settings where access to advanced diagnostic tools may be restricted.

## 5. Conclusions

The integration of EBUS-TBNA and mediastinoscopy in mediastinal staging represents a strategic approach to optimizing diagnostic accuracy in NSCLC. While EBUS-TBNA has largely replaced mediastinoscopy as the preferred first-line diagnostic tool due to its minimally invasive nature, lower complication rates, and cost-effectiveness, mediastinoscopy remains indispensable in select cases requiring posterior mediastinal lymph node assessment or larger biopsy samples for comprehensive histopathological and molecular analysis.

A sequential approach, where EBUS-TBNA is used initially and mediastinoscopy is reserved for cases requiring additional tissue sampling, has demonstrated superior staging accuracy compared to either technique alone. This stepwise integration not only reduces unnecessary surgeries and healthcare costs but also ensures patients receive the most appropriate treatment based on accurate staging results.

## Figures and Tables

**Figure 1 healthcare-13-01924-f001:**
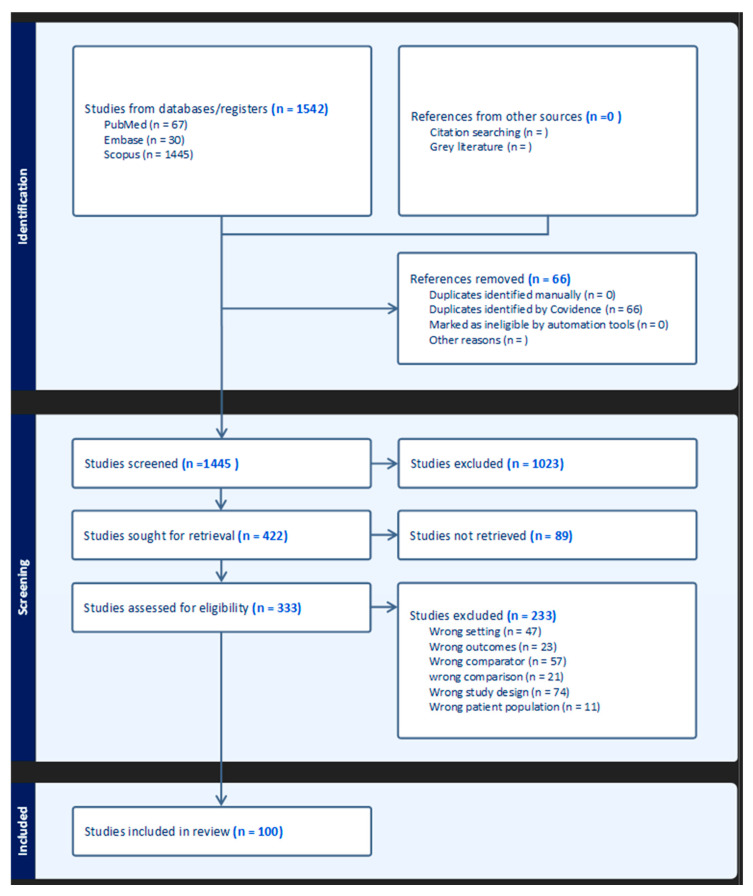
PRISMA diagram.

**Table 1 healthcare-13-01924-t001:** Comparative analysis of EBUS-TBNA and mediastinoscopy in NSCLC diagnosis.

Study	Year	Sensitivity (EBUS-TBNA)	Specificity (EBUS-TBNA)	Procedures (EBUS-TBNA)	Sensitivity (Mediastinoscopy)	Specificity (Mediastinoscopy)	Procedures (Mediastinoscopy)
Medford A. et al. (2009) [12]	2009	87	100	456	89	100	420
Annema JT et al. (2010) [13]	2010	85	100	273	86	100	260
Nakajima et al. (2011) [14]	2011	94	100	153	91	100	145
Wei B et al. (2012) [15]	2012	91	100	300	91	99	285
Harewood GC et al. (2019) [16]	2012	90	99	321	90	99	310
Verhagen AF et al. (2013) [17]	2013	86	99	312	88	99	300
Eapen GA et al. (2013) [18]	2015	94	100	487	89	100	470
Sehgal IS et al. (2016) [11]	2016	88	100	432	89	100	400
Kemp SV et al. (2020) [19]	2020	93	99	542	90	100	520

**Table 2 healthcare-13-01924-t002:** Diagnostic yield, sample adequacy, and molecular testing feasibility of EBUS-TBNA in NSCLC.

Study	Year	Modality	Diagnostic Yield (%)	Sample Adequacy (%)	Molecular Testing Feasibility (%)
Yasufuku et al. [22]	2011	EBUS-TBNA	94	100	100
Zieliński et al. [23]	2013	EBUS-TBNA	85	95	90
Nakajima et al. [24]	2012	EBUS-TBNA	92	98	95
Herth et al. [25]	2012	EBUS-TBNA	89	97	93
Oki et al. [26]	2011	EBUS-TBNA	91	96	94
Szlubowski et al. [23]	2013	EBUS-TBNA	87	94	92
Defranchi et al. [27]	2010	EBUS-TBNA	90	95	91
Evison et al. [28]	2014	EBUS-TBNA	88	96	93
Gelberg J et al. [29]	2014	EBUS-TBNA	86	94	92
Tournoy et al. [30]	2012	EBUS-TBNA	93	99	96

**Table 3 healthcare-13-01924-t003:** Comparison of risks and complications associated with EBUS-TBNA and mediastinoscopy.

Procedure	Common Risks (with Percentage)	Rare Complications	Morbidity Rate	Mortality Rate
EBUS-TBNA	Minor bleeding (1–2%), transient hypoxemia, vocal cord trauma	Pneumothorax (<1%), mediastinitis (<0.5%)	<2%	<0.01%
Mediastinoscopy	Bleeding (up to 1%), pneumothorax (1–3%), recurrent laryngeal nerve injury (1–2%)	Mediastinitis (<0.1%), wound infections	3–5%	<0.1%

**Table 4 healthcare-13-01924-t004:** Comparative analysis of EBUS-TBNA and mediastinoscopy—complications.

Study	Number of Procedures (EBUS-TBNA)	Number of Procedures (Mediastinoscopy)	Complications Rate (EBUS-TBNA)	Complications Rate (Mediastinoscopy)	Mortality Rate (EBUS-TBNA)	Mortality Rate (Mediastinoscopy)	Adverse Events (EBUS-TBNA)	Adverse Events (Mediastinoscopy)
Kemp SV et al. (2020) [43]	542	520	<2%	3–5%	<0.01%	<0.1%	Minimal	Moderate
Sehgal IS et al. (2019) [11]	500	480	<2%	4%	<0.01%	<0.1%	Minimal	Moderate
Evison M et al. (2018) [44]	450	400	1.50%	3.80%	<0.01%	<0.1%	Minimal	Moderate
Eapen GA et al. (2013) [18]	600	590	1.40%	4.20%	<0.01%	<0.1%	Minimal	Moderate
Verhagen AF et al. (2019) [45]	520	500	<2%	4%	<0.01%	<0.1%	Minimal	Moderate
Silvestri GA et al. (2018) [46]	430	410	1.60%	3.90%	<0.01%	<0.1%	Minimal	Moderate
Detterbeck FC et al. (2020) [47]	510	490	<2%	4%	<0.01%	<0.1%	Minimal	Moderate
Annema JT et al. (2019) [48]	490	480	<2%	4.10%	<0.01%	<0.1%	Minimal	Moderate
Rusch VW et al. (2020) [49]	550	540	1.80%	3.70%	<0.01%	<0.1%	Minimal	Moderate

**Table 5 healthcare-13-01924-t005:** Cost-effectiveness of EBUS-TBNA and mediastinoscopy in mediastinal staging of NSCLC.

Study	Procedure	Overall Cost per Procedure	Country
Steinhauser Motta JP et al. (2020) [55]	EBUS-TBNA vs. mediastinoscopy	EBUS-TBNA strategies were consistently less costly than mediastinoscopy across all analysed studies.	Systematic review encompassing multiple countries
Czarnecka-Kujawa K et al. (2018) [56]	EBUS-TBNA vs. mediastinoscopy	EBUS-TBNA staging had an incremental cost-effectiveness ratio (ICER) of CAD 26,000 per QALY; mediastinoscopy was dominated (less effective and more costly).	Canada
Sharples LD et al. (2012) [57]	Endosonography (EBUS-TBNA and EUS-FNA) followed by surgical staging vs. surgical staging alone	No significant difference in expected costs between strategies; endosonography had a 91% probability of being cost-effective at a willingness-to-pay threshold of GBP 30,000 per QALY.	United Kingdom
Søgaard R et al. (2013) [58]	PET-CT followed by EBUS-TBNA for positive findings vs. other strategies	PET-CT followed by EBUS-TBNA was the least expensive strategy; sending all patients directly to EBUS-TBNA also dominated other strategies.	Denmark
Evison M et al. (2018) [59]	EBUS-TBNA vs. mediastinoscopy	EBUS-TBNA was found to be more cost-effective compared to mediastinoscopy for mediastinal staging in lung cancer.	United Kingdom

**Table 6 healthcare-13-01924-t006:** Anaesthesia practices in EBUS-TBNA and mediastinoscopy.

Study	Year	Procedure	Anaesthesia Type	Key Findings
Casal et al. (2015) [68]	2015	EBUS-TBNA	General Anaesthesia vs. Moderate Sedation	No significant difference in diagnostic yield, but general anaesthesia increased procedure time.
Dal et al. (2015) [69]	2015	EBUS-TBNA	Ketamine–Midazolam vs. Ketamine–Propofol	Both sedation regimens were effective and safe with similar patient comfort.
Yarmus et al. (2013) [70]	2013	EBUS-TBNA	Moderate Sedation vs. General Anaesthesia	General anaesthesia had higher diagnostic yield, but patient selection was crucial.
Steinhauser Motta et al. (2022) [71]	2022	EBUS-TBNA vs. Mediastinoscopy	Not Specified	EBUS-TBNA was the least costly strategy for invasive mediastinal staging.
Kang et al. (2020) [72]	2020	EBUS-TBNA	Conscious Sedation (Midazolam and Meperidine)	Midazolam and meperidine were safe and effective for EBUS-TBNA sedation.
Harewood et al. (2010) [61]	2021	EBUS-TBNA	General Anaesthesia vs. Moderate Sedation	No significant difference in respiratory complications between sedation types.
Eapen et al. (2013) [18]	2013	EBUS-TBNA	Various Sedation Practices	Low complication rates were observed across sedation practices.
Um et al. (2015) [73]	2015	EBUS-TBNA	Conscious Sedation	High patient satisfaction and diagnostic yield with conscious sedation.
Aswanetmanee et al. (2016) [74]	2016	EBUS-TBNA	Various Sedation Practices	Moderate sedation and general anaesthesia were both viable options with differing recovery times.

## Data Availability

All articles included in this systematic review are publicly available through recognized online scientific databases such as PubMed, Scopus, and Web of Science. No unpublished data were used in this analysis.

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
