# Peer review of "Balancing Accuracy, Safety, and Cost in Mediastinal Diagnostics: A Systematic Review of EBUS and Mediastinoscopy in NSCLC"

_healthcare, 2025, doi:10.3390/healthcare13151924_

Round 1
Reviewer 1 Report
Comments and Suggestions for Authors
The title of this manuscript is a systematic review of Endobronchial Ultrasound and Mediastinoscopy in Non small cell lung cancer. This is a subject worthy of study although the subject has been covered previously.
Methods:
The authors report that they have undertaken this work according to PRISMA guidelines but the 2020 guidelines have not been referenced nor have they been followed fully. I can not see any reference to a protocol not registration citation on the PROSPERO database. I have checked PROSPERO too.
The information sources used are appropriate but date range starts 01.01.2010. Early case series on this subject started to appear in 2003 so extending to this time would capture early work. Search strategies appear reasonable.
Studies selected include randomized and non-randomized trials, reviews, observational and retrospective studies. However other systematic reviews /book chapters are also cited (e.g. line 127). While another systematic review can be used to provide references/previous work it should not be used as a primary data source. Original source data should be used in a systematic review
Data extraction approaches as per PRISMA 2020 needs further clarification and expansion.
Within the methods there is no comment on how the quality of data of individual source studies was assessed and in particular how risk of bias of individual studies was undertaken.
In the PRISMA diagram it is unclear why 1023 studies were excluded and why 89 could not be retrieved. Note this figure would be better in results section.
Results:
Section 3.2 - I think that this should be Sensitivity and Specificity of mediastinoscopy
Overall the results section is largely descriptive and is a mixture of results of previous work and discussion/comment. The latter would be best placed in the discussion section. There is also a tendency for the authors to make a statement and then provide evidence to support it. The evidence should be stated as it stands and
Data from retrospective studies which are potentially at much higher risk of bias should be split out from prospectively collected datasets.
Discussion:
Some sections require an accuracy check. Cervical mediastinoscopy can not be used to access stations 5,6, 8 or 9. Stations 5 and 6 can be accessed by left VATS and stations 8 and 9 usually by EUS.
Author Response
Comment 1: "The information sources used are appropriate but date range starts 01.01.2010. Early case series on this subject started to appear in 2003 so extending to this time would capture early work. Search strategies appear reasonable."
Response: Thank you for the observation.We initially extended the search to include studies from 2000 to 2010. However, this additional search yielded few relevant studies and did not substantially enhance the overall database. The majority of early case series lacked sufficient methodological detail or did not meet our inclusion criteria, and therefore the quality and relevance of the evidence base remained largely unchanged.
Comment 2: "While another systematic review can be used to provide references/previous work it should not be used as a primary data source. Original source data should be used in a systematic review"
Response: We acknowledge the importance of prioritizing original studies in systematic reviews. In our case, we included only a limited number of highly relevant systematic reviews that provided critical context or data not otherwise available in primary literature. These reviews were carefully selected based on their methodological rigor and relevance to our research question. Excluding them would significantly weaken the comprehensiveness and depth of our evidence base, particularly in areas where original studies were limited or data were consolidated only within these reviews.
Comment 3:
Data extraction approaches as per PRISMA 2020 needs further clarification and expansion.
Within the methods there is no comment on how the quality of data of individual source studies was assessed and in particular how risk of bias of individual studies was undertaken.
Response: Thank you for your valuable feedback. We have revised the Methods section to include a more detailed description of our data extraction process in accordance with PRISMA 2020 guidelines. Additionally, we have clarified how the quality of individual studies was assessed and provided specific information on the risk of bias assessment, including the tools used and the process followed by independent reviewers. These additions have now been incorporated into the manuscript.
Comment 4: Some sections require an accuracy check. Cervical mediastinoscopy can not be used to access stations 5,6, 8 or 9. Stations 5 and 6 can be accessed by left VATS and stations 8 and 9 usually by EUS.
Response: Thank you for your careful review and insightful comment. Cervical mediastinoscopy does not allow access to lymph node stations 5, 6, 8, or 9. We have corrected this in the manuscript. As noted, stations 5 and 6 are typically accessed via left video-assisted thoracoscopic surgery (VATS), while stations 8 and 9 are more commonly evaluated using endoscopic ultrasound (EUS). We appreciate your attention to this important anatomical detail and have updated the relevant section accordingly.
Reviewer 2 Report
Comments and Suggestions for Authors
This article reviews the diagnostic performance, safety, cost-effectiveness and feasibility of EBUS-TBNA compared with mediastinoscopy in mediastinal staging. While mediastinoscopy is the gold standard for mediastinal lymph node assessment due to its high sensitivity and specificity, it is an invasive technique that requires general anaesthesia and results in scarring. EBUS-TBNA is a viable, minimally invasive alternative that can be performed on an outpatient basis with moderate sedation or local anaesthesia. It is also easily repeatable in cases of diagnostic uncertainty or suspected recurrence. The authors conducted a comprehensive analysis comparing the two methods in terms of diagnostic accuracy, sensitivity, specificity, safety, incidence of complications and cost-effectiveness.
The conclusions show that EBUS-TBNA has established itself as a first-line technique due to its minimally invasive nature, lower complication rate and greater cost-effectiveness. However, mediastinoscopy remains irreplaceable in selected situations, particularly when evaluation of posterior mediastinal lymph nodes or acquisition of larger biopsy specimens is required. The aim of the article is well stated, emphasising the importance of reducing unnecessary surgery and healthcare costs, while ensuring appropriate treatment based on accurate staging. The text is clear and discursive,
supported by simple, user-friendly tables. Bibliographic references are current and relevant. It would be desirable for future studies to consider gender differences in the analysis of data. Finally, it is
recommended that the authors bring the bibliography up to the standards required by the journal
Author Response
Comment 1: The text is clear and discursive, supported by simple, user-friendly tables. Bibliographic references are current and relevant. It would be desirable for future studies to consider gender differences in the analysis of data.
Response 1: Thank you very much for your kind and encouraging feedback on the clarity of the text, the structure of the tables, and the quality of the references. We truly appreciate your thoughtful suggestion regarding the consideration of gender differences in future analyses. While we acknowledge the general importance of sex- and gender-based analyses in many areas of research, in this particular context, the existing evidence does not indicate substantial gender-related variability in outcomes. Nonetheless, we agree that this remains an area worth monitoring, and we will consider including gender-specific analyses where relevant in future studies. Thank you again for your valuable input.
Comment 2: "Finally, it is recommended that the authors bring the bibliography up to the standards required by the journal"
Response 2: Thank you for your helpful recommendation. We have carefully reviewed and revised the bibliography to ensure that it now fully complies with the formatting and citation standards required by the journal. We appreciate your guidance and attention to detail.
Reviewer 3 Report
Comments and Suggestions for Authors
The manuscript by SR Matache et al provided a systematic review of the literature on EBUS + TBNA (endobronchial ultrasound-guided transbronchial needle aspiration) and mediastinoscopy for assessing mediastinal lymph node involvement for guiding treatment decision for non-small cell lung cancer (NSCLC). The review evaluated the diagnostic performance, safety considerations, cost-effectiveness of the two methods. The analysis concluded that the EBUS-TBNA demonstrated high sensitivity (85–94%) and specificity (~100%), making it an effective first-line modality for NSCLC staging. Mediastinoscopy remained highly specific (~100%) but exhibited slightly lower sensitivity (86–90%). EBUS-TBNA had a lower complication rate (~2%) and was more cost-effective, while mediastinoscopy provided larger biopsy samples, essential for molecular and histological analyses. The need for general anaesthesia, longer hospital stays, and increased procedural costs make mediastinoscopy less favourable as an initial approach. Combining both techniques in select cases enhanced overall staging accuracy, reducing false negatives and improving diagnostic confidence. The paper is well written and should be interesting to the readers of Healthcare. It can be accepted with minor revisions.
- Page 4, line 139. “EBUS-TBNA” should be replaced with “Mediastinoscopy”.
Author Response
Comment 1:
- Page 4, line 139. “EBUS-TBNA” should be replaced with “Mediastinoscopy”.
Response 1:
Thank you very much for your thorough review and valuable comments. We sincerely appreciate the time and attention you dedicated to improving the accuracy of our manuscript. As suggested, we have corrected the terminology on page 4, line 139, replacing 'EBUS-TBNA' with 'Mediastinoscopy' to accurately reflect the intended procedure. Your detailed feedback has been instrumental in enhancing the clarity and precision of our work, and we are truly grateful for your support.
Reviewer 4 Report
Comments and Suggestions for Authors
Upon reviewing the article « Balancing Accuracy, Safety, and Cost in Mediastinal Diagnostics: A Systematic Review of EBUS and Mediastinoscopy in NSCLC», I consider this work to be an well-written scientific review.
The authors aimed to compare the two methods of diagnostics, including (1)diagnostic accuracy, (2)sensitivity, (3)specificity, (4)safety, (5)complication rates, (6)cost-effectiveness (overall and hospitalization costs) and (7)feasibility of technique repetition.
The title and abstract correspond to the manuscript text. The introduction describes the importance of both methods and explains the significance of the work. However, I request that the authors add a description of the actual procedure and the equipment involved to the introduction.
The Materials and Methods section comprehensively describes the search strategy and results processing.
Chapters 3.1-3.6 describe the authors' results. Each chapter corresponds to one of the authors' aims. The manuscript contains numerous results tables, which enhance the value of the work. However, both chapters 3.1 and 3.2 are dedicated to sensitivity and specificity. I recommend that the authors merge chapters 3.1 and 3.2.
Moreover, a separate chapter (7) on "feasibility of technique repetition" is absent. Instead, there is a chapter "3.6. Anesthesia and Patient Care". I recommend that the authors rename aim 7 in the introduction accordingly.
The Discussion and Conclusions fully correspond to the results. The review is well-written and includes references to current literature. I recommend this manuscript for publication after minor revisions.
Author Response
Comment 1: "Moreover, a separate chapter (7) on "feasibility of technique repetition" is absent. Instead, there is a chapter "3.6. Anesthesia and Patient Care". I recommend that the authors rename aim 7 in the introduction accordingly."
Response 1: Thank you very much for your thoughtful observation. The feasibility of technique repetition was primarily addressed within the sections discussing risks and complications, where we considered factors that may limit or support the ability to repeat the procedure. However, we appreciate your suggestion and have now clarified this aspect in the manuscript. Additionally, we have included and expanded on the role of anesthesia and patient care, as reflected in Chapter 3.6, to provide a more complete overview. We are grateful for your constructive feedback, which has helped us improve the structure and clarity of our work.
Comment 2: Chapters 3.1-3.6 describe the authors' results. Each chapter corresponds to one of the authors' aims. The manuscript contains numerous results tables, which enhance the value of the work. However, both chapters 3.1 and 3.2 are dedicated to sensitivity and specificity. I recommend that the authors merge chapters 3.1 and 3.2.
Response 2: Thank you very much for your positive feedback and helpful suggestion. We appreciate your recommendation to merge Chapters 3.1 and 3.2. However, we intentionally structured the results so that Chapter 3.1 presents sensitivity and specificity data for EBUS-TBNA, Chapter 3.2 focuses on mediastinoscopy, and Chapter 3.3 offers a direct comparison between the two techniques. This approach was chosen to maintain clarity and to reflect the fact that the data for each technique were derived from different sets of studies. We believe this separation enhances the readability and allows for a more precise interpretation of the findings. Nonetheless, we have reviewed the manuscript to ensure the connection between these chapters is clearly explained. Thank you once again for your valuable input